# Multidimensional Prognostic Index and Outcomes in Older Patients Undergoing Transcatheter Aortic Valve Implantation: Survival of the Fittest

**DOI:** 10.3390/jcm10163529

**Published:** 2021-08-11

**Authors:** Jeannette A. Goudzwaard, Sadhna Chotkan, Marjo J. A. G. De Ronde-Tillmans, Mattie J. Lenzen, Maarten P. H. van Wiechen, Joris F. W. Ooms, Harmke A. Polinder-Bos, Madelon de Beer-Leentfaar, Nicolas M. Van Mieghem, Joost Daemen, Alberto Pilotto, Peter P. T. de Jaegere, Francesco U. S. Mattace-Raso

**Affiliations:** 1Section of Geriatrics, Department of Internal Medicine, Erasmus MC University Medical Center, 3015 GD Rotterdam, The Netherlands; chotkan.sa@gmail.com (S.C.); h.polinder-bos@erasmusmc.nl (H.A.P.-B.); m.debeer-leentfaar@erasmusmc.nl (M.d.B.-L.); f.mattaceraso@erasmusmc.nl (F.U.S.M.-R.); 2Department of Cardiology, Thoraxcenter, Erasmus MC University Medical Center, 3015 GD Rotterdam, The Netherlands; m.j.a.g.deronde@erasmusmc.nl (M.J.A.G.D.R.-T.); m.lenzen@erasmusmc.nl (M.J.L.); m.vanwiechen@erasmusmc.nl (M.P.H.v.W.); j.f.w.ooms@erasmusmc.nl (J.F.W.O.); n.vanmieghem@erasmusmc.nl (N.M.V.M.); j.daemen@erasmusmc.nl (J.D.); p.dejaegere@erasmusmc.nl (P.P.T.d.J.); 3Department of Geriatric Care, Orthogeriatrics and Rehabilitation, E.O. Galliera Hospital, 16128 Genoa, Italy; alberto.pilotto@galliera.it; 4Department of Interdisciplinary Medicine, University of Bari, 70121 Bari, Italy

**Keywords:** aortic stenosis (AS), transcatheter aortic valve implantation (TAVI), multidimensional prognostic index, outcome, mortality, frailty

## Abstract

Selecting patients with a high chance of endured benefit from transcatheter aortic valve implantation (TAVI) is becoming relevant with changing indications and increasing number of TAVI being performed. The aim of our study was to investigate the association of the multidimensional prognostic index (MPI) based on a comprehensive geriatric assessment (CGA) on survival. The TAVI Care & Cure program is a prospective, observational registry of patients referred for TAVI at the Erasmus MC University Medical Center. Consecutive patients who underwent a complete CGA and TAVI were included. CGA components were used to calculate the MPI score. The impact of the MPI score on survival was evaluated using Cox regression. Furthermore, 376 patients were included, 143 (38.0%) patients belonged to the MPI-1 group and 233 (61.9%) patients to the MPI-2–3 group. After 3 years, 14.9% of the patients in the MPI-1 group and 30.5% of the patients in the MPI-2–3 group died (*p* = 0.001). Patients in MPI-1 had increased chances of overall survival in comparison with patients in MPI group 2–3 Hazard Ratio (HR) 0.57, (95% Confidence Interval (CI) 0.33–0.98)). In this study we found that the MPI tool could be useful to assess frailty and to predict which patient will have a higher chance of enduring benefit from a TAVI procedure.

## 1. Introduction

The indications for transcatheter aortic valve implantation (TAVI) for treating symptomatic aortic stenosis are expanding from older patients who are frail and have high surgical risk to patients with low surgical risk [1,2,3,4]. With the rapid uptake of TAVIs being performed [5], selection of patients who can benefit from TAVI is becoming more relevant. Performing a comprehensive geriatric assessment (CGA) is a growing routine practice and the impact of frailty status of patients on outcomes after TAVI is increasingly described [6,7,8,9]. The multidimensional prognostic index (MPI) is based on a CGA and has been shown to predict mortality in older patients with acute and chronic conditions [10], including cardiovascular diseases, i.e., heart failure [11] and acute myocardial infarction [12]. In a large, multicenter, longitudinal study, the MPI has shown to be predictive of mortality and negative health outcomes in older, hospitalized patients [13]. Recent studies in relatively small groups of patients, have suggested that the MPI can predict death and stroke for up to three months after TAVI, and mortality for up to one year after TAVI [14,15,16]. Since the population of patients qualifying for a TAVI is changing and growing, late outcomes are also becoming important. There has been limited research on the effect of frailty and long-term survival after TAVI [17,18]. The aim of this study was therefore to investigate the association between the MPI and survival at 1 and 3 years in older patients undergoing TAVI.

## 2. Materials and Methods

### 2.1. Study Population

The study population consists of patients who underwent TAVI (November 2013–July 2018) within the framework of the TAVI Care & Cure program [19]. The TAVI Care & Cure program is a collaboration between the departments of geriatrics and interventional cardiology to optimize care for older patients. Patients referred for severe aortic valve stenosis are seen by the interventional cardiologist for a cardiac assessment, followed by a consultation by the geriatrician to complete a comprehensive geriatric assessment (CGA). Predefined cardiovascular and non-cardiovascular characteristics, procedural and postoperative data of all patients referred for and treated with TAVI were collected [19]. There were no specific exclusion criteria. Treatment decision and strategy were decided during the multidisciplinary heart team meeting (interventional cardiologists, cardiac surgeons, anesthesiologists, geriatricians and a TAVI-nurse coordinator) [19,20,21]. The study was approved by the Medical Ethics Committee of the Erasmus MC University Medical Center and was conducted according to the Helsinki Declaration. All participants provided written informed consent.

### 2.2. Cardiology Assessment

Cardiology assessment included determining symptoms using the New York Heart Association (NYHA) classification and the Canadian Cardiovascular Society (CCS) grading of angina pectoris, medical history, physical examination, laboratory assessment and electrocardiogram [19]. Echocardiography, coronary angiography and multislice computed tomography (MSCT) were examined to evaluate the condition of the aortic valve and to determine access site [22].

### 2.3. Comprehensive Geriatric Assessment

The MPI is based on a standardized CGA and includes eight domains [10]. We calculated the MPI as described in previous studies [11,14,23], with some modifications based on availability of data. We used five of the eight original MPI domains: Activities of Daily Living (ADL) [24], instrumental activities of daily living (IADL) [25], cumulative illness rating scale and comorbidity index (CIRS-CI) [26], the number of medications and social support network. For the cognition, malnutrition and pressure risk domain scores we used different, validated instruments, which have been used previously in calculating the MPI [23]. For the cognition domain we used the mini mental state examination (MMSE) [27], for the domain malnutrition we used the malnutrition screening tool (MUST) [28] and for the pressure risk domain score we used the Waterlow score [29]. The cumulative illness rating scale for geriatrics (CIRS-G) measures chronic medical illness burden while taking into account the severity of the chronic disease across 14 items representing individual body systems. The cumulative final score can vary theoretically from 0 to 56. The severity index is calculated by dividing the total score through the total number of categories endorsed [26].

For each of the eight domains, a three-level score was assigned with score 0 indicating no problem, score 0.5 indicating a minor problem and score 1 indicating a severe problem, as established in previous studies [10,14,23]. The categorization of each domain can be found in Table 1. The sum of all domain values is then divided by 8 to obtain the final MPI score ranging between 0 and 1. Since our aim was to verify the effectiveness of the previously established index in this specific cohort, we used the previously defined cut-off points for the risk of mortality: MPI-1 score 0–0.33, indicating low risk, MPI-2 score 0.34–0.66, indicating medium risk and MPI-3 score 0.67–1.0, indicating high risk [10].

As complementary functional tests we used two validated tests for mobility: the 5 meter gait speed test and the timed up and go test. A gait speed of ≤1 m/s is suspect of moderate or severe limitation of mobility [30]. Slowness was evaluated with the timed up and go test. A timed up and go test of ≥20 s confirms moderate or severe limitation of mobility [31].

### 2.4. Outcome Measures

Primary outcomes were survival at 1 year and 3 years after TAVI. Secondary outcomes included vascular complications (in-hospital life-threatening or major bleeding or other vascular complications), in-hospital stroke, infection, delirium and 30-day mortality. Delirium was defined according to the Diagnostic and Statistical Manual of Mental Disorders, Fourth Edition (DSM-IV). Life-threatening or major bleeding, vascular complications and stroke were assessed according to the guidelines of the Valve Academic Research Consortium [32]. Procedural outcomes and mortality were assessed prospectively by consulting medical files and the Dutch Civil Registry.

### 2.5. Statistical Analysis

Categorical variables are presented as numbers and corresponding percentages and differences between MPI groups with the chi-square or Fisher’s exact tests as appropriate. Continuous variables are expressed as means ± SD or median values with corresponding interquartile ranges (IQR) and differences between MPI groups were compared using the independent t-test or its non-parametric equivalents, respectively. A Cox regression analysis was performed for the primary outcome survival. Hazard ratios (HR) and corresponding 95% confidence intervals (CI) were computed. Univariate analyses were performed, every variable with a *p* value < 0.10 was entered in the multivariate regression model. Variables in the multivariate regression analysis included: age, sex, MPI score, diabetes mellitus, limitation of mobility (5MGST), limitation of mobility (TUGT), reduced grip strength, logistic Euroscore, STS score and post procedural stroke. A logistic univariate regression analysis was performed for secondary outcome measures. *p* value of 0.05 was considered statistically significant. Data was analyzed with statistic program IBM Statistical Package for Social Science for Windows version 25, Rotterdam, The Netherlands (SPSS).

## 3. Results

### 3.1. Patient Characteristics

In total 895 patients underwent TAVI. Within this group, 376 patients completed baseline CGA and were included in this study. According to the MPI score, 143 (38.0%) patients belonged to the MPI-1 group, 221 (58.8%) patients to the MPI-2 group and 12 (3.2%) to the MPI-3 group (Table 2). As only 12 patients belonged to the MPI-3 group we combined this group with the MPI-2 group for further analyses. The baseline characteristics of the MPI group 1 and MPI group 2–3 are shown in Table 2. Patients in the MPI-2–3 group were older (82.0 ± 6.4 vs. 80.7 ± 5.7, *p* = 0.04) than patients in the MPI-1 group and less men belonged to MPI-2–3 group (38.2% vs. 68.1%, *p* < 0.001). Hypertension (82.8% vs. 71.6%, *p* = 0.013), diabetes mellitus (40.9% vs. 19.9%, *p* < 0.001), previous stroke (21.1% vs. 14.9%, *p* = 0.028) and renal dysfunction (49.4% vs. 33.3%, *p* = 0.002) were more prevalent in the MPI-2–3 group compared to the MPI-1 group. The mean logistic Euroscore was 16.8 (±11.0)%, with 111 (29.7%) patients considered to have a high surgical risk according to a logistic Euroscore ≥ 20%. Mean MPI score was 0.39 (±0.14) points.

### 3.2. Primary Outcomes

One year after TAVI, 87% of the total study population survived. The one-year survival rate in the MPI-1 group was 92% compared to 84% in the MPI-2–3 group (*p* = 0.018). In multivariate Cox regression analysis, the presence of renal dysfunction (*p* = 0.004) and limitation of mobility (5MGST) (*p* = 0.01) were associated with mortality one year after TAVI.

Three years after TAVI, 85.1% of patients belonging to MPI-1 group compared to 69.5% in the MPI-2–3 group were still alive (*p* = 0.001). In multivariate logistic regression analysis, belonging to the MPI group 2–3 was associated with mortality at 3 years after TAVI (HR 1.99 95% CI 1.13–3.50) (Figure 1).

Other factors associated with 3-year mortality were the presence of renal dysfunction (HR 1.85, 95% CI 1.19–2.87), reduced gait speed (HR 2.05, 95% CI 1.49–3.65) and the occurrence of post-procedural stroke (HR 2.96, 95% CI 1.26–6.99). The presence of hypercholesterolemia (HR 0.52, 95% CI 0.33–0.81) was associated with reduced overall mortality (Table 3).

Within the MPI-2–3 group women (HR 2.09 95% CI 1.230–3.562), patients with renal dysfunction before TAVI (HR 2.03 95% CI 1.21–3.40) and limitation of mobility (5MGST) (HR 2.00 95% CI 1.02–3.93) had the highest risk of 3-year mortality (Table 4).

### 3.3. Secondary Outcomes

There were 56 (14.9%) vascular complications in the 30 days after the procedure. Patients with a high risk MPI score (MPI score 2 or 3) had a higher risk of vascular complications in comparison to those in the MPI-1 group (OR 2.00, 95% CI 1.05–3.80, *p* = 0.04) in univariate regression analysis. When adjusted for age and sex, estimates were no longer statistically significant (OR 1.81, 95% CI 0.92–3.53). There were no differences found between the MPI groups for the other procedural outcomes (Table A1). Thirteen (3.5%) patients died within 30 days after the procedure; 2 (0.5%) within MPI-1 group and 11 (3.0%) patients within MPI-2–3 group (*p* = 0.26).

## 4. Discussion

In this study we found that the MPI tool is useful to assess frailty and could be able to predict survival after TAVI. One- and three-year survival rates were 92% and 85% in patients belonging to MPI-1, corresponding survival rates were 85% and 69.5% in patients belonging to MPI-2–3, respectively. Patients in MPI-1 had a 50% higher chance of long-term survival in comparison with patients in MPI-2–3.

The selection of older patients who will benefit the most from TAVI still remains a challenge. This study shows that the MPI tool could be helpful in the decision-making process. The traditional risk scores commonly used to predict surgical mortality (e.g., logistic Euroscore or the Society of Thoracic Surgeons (STS)) are not sufficient to predict mortality for TAVI procedure in high-risk patients of 80 years and older with substantial comorbidity [33,34]. The incorporation of frailty can have an additional prognostic role as a geriatric biomarker, as studies have shown that frailty is a predictive factor for negative health outcomes after TAVI [6,7,9]. Frailty is defined as a state of reduced physical, cognitive and social functioning, resulting in a reduction of reserve capacity for dealing with stressors [35]. Several conceptual models of frailty have been described; a general agreement exists, however, on the concept that frailty is a multidimensional condition, with physical and psychosocial factors playing a part in its development [35]. Accordingly, the use of the CGA, i.e., a multidimensional diagnostic process for evaluating clinical, functional, cognitive, nutritional and social characteristics of individuals, has been suggested as a clinically useful tool to guarantee a multidimensional approach to frailty [36], even in older patients with aortic stenosis [37]. The incorporation of CGA measures into clinical assessment of patients opting for TAVI procedure is becoming more routine in a growing number of centers [15,19]. The MPI is derived from a standard CGA [10] and has shown to predict mortality in older hospitalized patients, including patients with heart failure, acute myocardial infarction and transient ischemic attack [11,13,23,38]. The value of the MPI in predicting outcomes in TAVI patients has previously been investigated. In a study performed on behalf of the MPI_AGE Project, investigators found that CGA based on the MPI tool predicted prognosis in older patients undergoing TAVI procedure where mortality rate was significantly different between MPI groups at six and twelve months [14]. The MPI tool was also used in a multi-center TAVI registry and found that the MPI showed value for predicting the likelihood of death and a combination of either death, fatal stroke, or both, by one year after TAVI [16]. However, the number of patients included in previous studies was relatively small (116 patients and 71 patients, respectively). To the best of our knowledge this is the first study that investigated the role of the MPI tool for predicting long-term outcomes in a large population of older patients undergoing TAVI.

The incorporation of information derived from a CGA can predict frailty in patients. The assessment of frailty in older patients with comorbidities can give a clear vision of the individual reserve capacity defining the somatic, cognitive and functional situation secondary to the damage due to chronic and intercurrent diseases and the individual capacity to react to external stressors. With growing numbers of TAVI being performed and the growing amount of evidence that being frail is not only predictive of reduced survival, but also on other relevant outcomes such as periprocedural outcomes [39] and health-related quality of life [40], the incorporation of a CGA-based tool can help assess frailty and therefore aid in selecting patients who are less frail with a higher chance of endured benefit from TAVI. When decisions are made concerning therapeutic interventions, the estimation of patient survival is crucial to assess the balance of benefits and risks of performing TAVI.

This study has several limitations. First, results should be interpreted within the framework of a single center and therefore cannot be extrapolated to other groups of patients. Second, the MPI tool investigated in the present study is a somewhat modified version of the original. However, the same modifications have been used in other studies [23], with the same range of predictive value as the original MPI. Third, although the sample size of the complete population was relatively large, there was a low sample size in the MPI-3 group (3.2%), therefore we could not draw any conclusions about the prognosis after TAVI in this specific group.

Further research should focus on the use of the MPI and other geriatric outcomes such as functionality and post-operative HRQoL, since for this specific population survival may be of less importance than improving functional status and quality of life.

In conclusion, in this study we found that the MPI tool is useful to assess frailty and predicts survival after TAVI. We found that in patients eligible for TAVI, 92% of patients belonging to the more vital group (MPI-1) are still alive one year after TAVI and 85% three years after the procedure in comparison to 84% and 69%, respectively, in the frailer MPI-2–3 group. The MPI tool could be useful to assess frailty and to predict which patients will have a higher change of enduring benefit from a TAVI procedure.

## Figures and Tables

**Figure 1 jcm-10-03529-f001:**
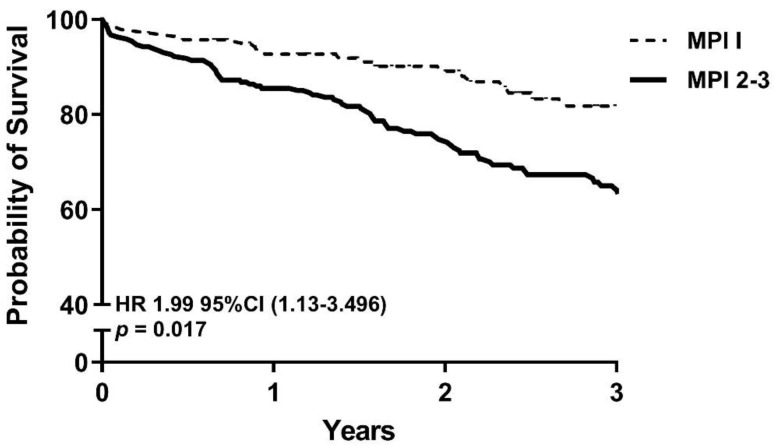
Cox regression survival curve stratified by MPI groups. Model adjusted for age, sex, diabetes mellitus, hypercholesterolemia, peripheral artery disease, renal dysfunction, limitation of mobility, logistic Euroscore, post-procedural stroke and delirium. Abbreviations used: MPI, Multiple Prognostic Index; HR, Hazard Ratio; CI, Confidence Interval.

**Table 1 jcm-10-03529-t001:** Multidimensional prognostic index score assigned to each domain based on severity of the problem.

Assessment	No Problem (Value = 0)	Minor Problem (Value = 0.5)	Severe Problem (Value = 1)
ADL	0	1–6	7–12
IADL	0–1	2–7	8–14
MMSE	28–30	25–27	0–24
CIRS-CI	0	1–2	≥3
MUST	0	1	≥2
Waterlow score	3–9	10–14	15–45
Number of medications	0–3	4–6	≥7
Social support network	Living with family	Institutionalized	Living alone

Abbreviations used: ADL, activities of daily living; IADL, instrumental activities of daily living; MMSE, mini mental state examination; CIRS-CI, cumulative illness rating scale and comorbidity index; MUST, malnutrition screening tool.

**Table 2 jcm-10-03529-t002:** Baseline patient characteristics (*n* = 376).

Characteristic	Total (377)	MPI-1 *N* = 141	MPI-2–3 *N* = 233	*p* Value
Age (y)	81.54 (±6.1)	80.68 (±5.7)	82.03 (±6.4)	0.040
Men (%)	189 (49.7%)	96 (68.1%)	89 (38.2%)	<0.001
BMI (kg/m^2^)	27.2 (±4.8)	27.35 (±4.25)	27.14 (±5.20)	0.697
Cardiovascular risk factors				
Hypertension (%)	294 (78.2%)	101 (71.6%)	192 (82.8%)	0.013
Hypercholesterolemia (%)	231 (61.4%)	83 (58.9%)	147 (63.9%)	0.378
Diabetes mellitus (%)	124 (33.1%)	28 (19.9%)	95 (40.9%)	<0.001
Current smoker (%)	30 (8.0%)	13 (9.2%)	17 (7.3%)	0.558
Comorbidities				
Previous myocardial infarction (%)	73 (19.4%)	27 (19.1%)	46 (19.7%)	1.00
Previous stroke (%)	82 (21.8%)	21 (14.9%)	49 (21.1%)	0.028
COPD (%)	82 (21.8%)	35 (24.8%)	47 (20.4%)	0.367
Renal dysfunction (%)	162 (43.5%)	47 (33.3%)	115 (49.4%)	0.002
CIRS index	1.91 (±0.27)	1.84 (± 0.26)	1.96 (±0.26)	<0.001
Symptoms				
NYHA Class 3 or 4 (%)	244 (64.9%)	73 (51.8%)	171 (73.4%)	<0.001
Angina CCS classification 3 or 4 (%)	43 (11.4%)	13 (9.4%)	30 (13.2%)	0.318
Vertigo (%)	143 (41.4%)	53 (40.5%)	90 (42.5%)	0.736
Echocardiography				
AV area (cm^2^)	0.76 (± 0.24)	0.8 (±0.23)	0.74 (±0.25)	0.052
Peak AoV, (m/s)	4.0 (± 0.70)	4.0 (±0.69)	4.0 (±0.71)	0.923
Cardiovascular risk scores				
Logistic Euroscore	16.82 (±11.03)	14.80 (±9.55)	18.02 (±11.70)	0.006
STS score	5.47 (±3.03)	4.29 (±2.19)	6.17 (±3.25)	0.228
CGA domains				
Cognitive impairment probable (%)	111 (29.5%)	20 (14.2%)	91 (39.1%)	<0.001
Malnutrition probable (%)	41 (10.9%)	5 (3.5%)	36 (15.5%)	<0.001
Limitation of mobility, TUGT (%)	48 (12.8%)	5 (3.9%)	43 (21.6%)	<0.001
Limitation of mobility, 5MGS (%)	219 (58.2%)	61 (48.8%)	158 (77.5%)	<0.001
Reduced muscle strength, male (%)	65 (17.3%)	24 (17%)	39 (16.7%)	0.004
Reduces muscle strength, female (%)	100 (26.6%)	17 (12.1%)	83 (35.6%)	0.016
Limitation in ADL activity (%)	111 (29.5%)	9 (6.4%)	102 (43.8%)	<0.001
Limitation in IADL activity (%)	200 (53.2%)	35 (24.8%)	165 (70.8%)	<0.001

Abbreviations used: BMI, body mass index; COPD, chronic obstructive pulmonary disease; CIRS, cumulative illness rating scale; NYHA, New York Heart Association; CCS, Canadian Cardiovascular Society; AoV, aortic valve; STS, Society for Thoracic Surgeons; TUGT, timed up and go test; 5MGS, 5 meter gait speed; ADL, activities of daily living; IADL, instrumental activities of daily living.

**Table 3 jcm-10-03529-t003:** Multivariable Cox analysis for primary outcome 3-year mortality.

Variable	HR	95% CI	*p*-Value
Age	0.99	0.95–1.03	0.51
Sex (men)	0.63	0.84–1.03	0.07
MPI-1 vs. MPI-2–3	1.99	1.13–3.50	0.02
Diabetes mellitus	1.16	0.71–1.90	0.56
Hypercholesterolemia	0.80	0.50–0.32	0.003
Peripheral artery disease	1.34	0.84–2.12	0.23
Renal dysfunction	1.85	1.19–2.87	0.006
Limitation of mobility (5MGST)	2.09	1.15–3.65	0.02
Logistic Euroscore	1.01	0.99–1.03	0.43
Post procedural stroke	2.96	1.26–6.99	0.01
Delirium	1.22	0.71–2.10	0.48

Abbreviations used: MPI, multidimensional prognostic index; 5MGS, 5 meter gait speed.

**Table 4 jcm-10-03529-t004:** Multivariable Cox analysis for 3-year mortality in the MPI-2–3 group.

Variable	HR	95% CI	*p*-Value
Age	0.99	0.95–1.04	0.66
Sex (women)	2.09	1.23–3.56	0.006
Diabetes mellitus	0.88	0.51–1.50	0.63
Hypercholesterolemia	0.47	0.28–0.80	0.005
Peripheral artery disease	1.42	0.82–2.46	0.21
Renal dysfunction	1.98	1.18–3.33	0.01
Limitation of mobility (5MGST)	1.19	0.97–3.79	0.06
Logistic Euroscore	1.01	0.98–1.03	0.66
Post procedural stroke	3.98	1.59–10.05	0.003
Delirium	0.97	0.52–1.81	0.93

Abbreviations used: MPI, multidimensional prognostic index; 5MGS, 5 meter gait speed.

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
