# Peer review of "Multidimensional Prognostic Index and Outcomes in Older Patients Undergoing Transcatheter Aortic Valve Implantation: Survival of the Fittest"

_jcm, 2021, doi:10.3390/jcm10163529_

Round 1

Reviewer 1 Report

Excellent manuscript.

It brings in another dimension, if 3 year survival is 70% even in the frail - I think I would still perform TAVI. Perhaps if MPI 1 we would certainly push harder to do TAVI , but if MP2/3 would we not do it? I think I still would - definitely worth thinking about, but I absolute agree it is essential that we continue to learn which patients fair best.

Author Response

Response to Reviewer 1 

First of all, I would like to thank the reviewer for taking the time and effort to read this paper. In a response to your review I provided a point-by-point response to all the remarks. 

It brings in another dimension, if 3 year survival is 70% even in the frail - I think I would still perform TAVI. Perhaps if MPI 1 we would certainly push harder to do TAVI , but if MP2/3 would we not do it? I think I still would - definitely worth thinking about, but I absolute agree it is essential that we continue to learn which patients fair best.

Thank you for your remark. And I fully agree that even in frail patients, TAVI offers opportunities for those who previously could not have been treated by means of surgical aortic valve replacement. The challenge will indeed be that within this frail population, we have to select those patients who belong to the 70% that will survive for a substantial amount of years. 

Reviewer 2 Report

The authors investigated the association of the multidimensional prognostic index on TAVI survival. Given that frailty is often included in the heart team's decision but is never qualified, it is a very exciting study. I have only a few comments:

  • How long does the MPI survey take and can this be routinely carried out in everyday clinical practice?
  • MPI seems to be a useful tool to predict survival after TAVI. However, you cannot conclude, which patient will have a higher change of enduring benefit from a TAVI procedure. To postulate this, you have to compare cohorts (with the same MPI) receiving TAVI and a conservative approach. Therefore, please clarify your conclusion.
  • Do you have an explanation, why hypercholesterolemia was associatedt with reduced overall mortality?
  • There are minor typos.

Author Response

Response to reviewer 2. 

First of all, I would like to thank the reviewer for taking the time and effort to read this paper. In a response to your review I provided a point-by-point response to all the remarks. 

'The authors investigated the association of the multidimensional prognostic index on TAVI survival. Given that frailty is often included in the heart team's decision but is never qualified, it is a very exciting study. I have only a few comments.'

How long does the MPI survey take and can this be routinely carried out in everyday clinical practice?

The MPI is overall based on components which are standard procedure within a Comprehensive Geriatric Assessment (CGA). This makes the MPI a tool which can be implemented in everyday clinical practice.  The only component that is not routinely used in a CGA is the Cumulative Illness Rating Scale for Geriatrics (CIRS-G). If trained, this scale can be finished in 5-10 minutes. In everyday practice, as part of the performed CGA, depending on experience the MPI can take between the 15 and 20 minutes.

MPI seems to be a useful tool to predict survival after TAVI. However, you cannot conclude, which patient will have a higher change of enduring benefit from a TAVI procedure. To postulate this, you have to compare cohorts (with the same MPI) receiving TAVI and a conservative approach. Therefore, please clarify your conclusion.

Thank you for this remark. In this study we studied survival rates and thereby endured benefit of TAVI in non frail (MPI I patients) versus frail patients (MPI 2-3). Previous studies has shown that the natural course of symptomatic aortic valve stenosis leads to high rates of mortality (approximately 50% in the first 2 years after symptoms appear), this includes patients that are treated conservative. The conservative treated patients are overall patients that are classified as inoperable or with an unacceptable high surgical risk. These patients would all belong to the MPI 2-3 group. TAVI has become the standard treatment procedure for these patients, but even though TAVI is a safe and effective procedure for older patients with high surgical risk, there remains a group of patients that even a minimally invasive interventional therapy could be harmful for them. In an older population, mostly with substantial comorbidity, selecting within this population who will benefit from TAVI is crucial. With the extensive studies that have been performed in the early days of TAVI showing that conservative treatment is inferior regarding survival rates, studies now should be performed to be aiding decision making regarding the older patients with or without frailty characteristics. A study with comparing cohorts including the MPI I group, with patients that would be considered vital, seems unethical considering the mortality rates in the conservative treated patients.

Since the study is performed in patients who were eligible for TAVI, we have adjusted the conclusion as follows:

In conclusion, in this study we found that the MPI tool is useful to assess frailty and predicts survival after TAVI. We found that, in patients eligible for TAVI, 92% of patients belonging to the more vital group (MPI-1) are still alive one year after TAVI and 85% 3 years after the procedure in comparison to 84% and  69 % respectively in the frailer MPI 2-3 group. The MPI tool could be useful to assess frailty and to predict which patient will have a higher change of enduring benefit from a TAVI procedure.

Do you have an explanation, why hypercholesterolemia was associated with reduced overall mortality?

Thank you for this question. There is literature that shows that hypercholesterolemia becomes less of a risk factor for all cause and cardiovascular mortality with increasing age. A study of Ravnskov et al, found that high LDL-cholesterol is inversely associated with mortality in most people over 60 years¹. A meta-analysis performed by the Prospective Studies Collaboration found in their analysis that the risk of hypercholesterolemia and cardiovascular death decreased with increasing age and became minimal after the age of 80 years².  In our study the mean age of the study population is 81.54 (±6.1) years, in light of the mentioned papers, the higher age of our study population could be the explanation for this result.

¹Ravnskov U, Diamond D,  Hama R, et al. Lack of association or an inverse association between low-density-lipoprotein cholesterol and mortality in the elderly: a systematic review. BMJ Open 2016;6:e010401. doi:10.1136/bmjopen-2015-010401

²Prospective Studies Collaboration. Lewington S, Whitlock G, Clarke R, et al. Blood cholesterol and vascular mortality by age, sex, and blood pressure: a meta-analysis of individual data from 61 prospective studies with 55,000 vascular deaths. Lancet 2007;370:1829–39. doi:10.1016/S0140-6736(07)61778-4

There are minor typos

Thank you for the remark, the manuscript has been checked and revised.

Round 2

Reviewer 2 Report

I would like to thank the authors for improving the manuscript. I think it is acceptable in the present form.